# Mitochondrial Processing Peptidases—Structure, Function and the Role in Human Diseases

**DOI:** 10.3390/ijms23031297

**Published:** 2022-01-24

**Authors:** Nina Kunová, Henrieta Havalová, Gabriela Ondrovičová, Barbora Stojkovičová, Jacob A. Bauer, Vladena Bauerová-Hlinková, Vladimir Pevala, Eva Kutejová

**Affiliations:** Department of Biochemistry and Protein Structure, Institute of Molecular Biology, Slovak Academy of Sciences, Dúbravská Cesta 21, 845 51 Bratislava, Slovakia; henrieta.havalova@savba.sk (H.H.); gabriela.ondrovicova@savba.sk (G.O.); barbora.keresztesova@savba.sk (B.S.); jacob.bauer@savba.sk (J.A.B.); vladena.bauerova@savba.sk (V.B.-H.); vladimir.pevala@savba.sk (V.P.)

**Keywords:** mitochondrial processing peptidases, MPP, MIP, IMP, mitochondrial rhomboid protease, mitochondrial disease

## Abstract

Mitochondrial proteins are encoded by both nuclear and mitochondrial DNA. While some of the essential subunits of the oxidative phosphorylation (OXPHOS) complexes responsible for cellular ATP production are synthesized directly in the mitochondria, most mitochondrial proteins are first translated in the cytosol and then imported into the organelle using a sophisticated transport system. These proteins are directed mainly by targeting presequences at their N-termini. These presequences need to be cleaved to allow the proper folding and assembly of the pre-proteins into functional protein complexes. In the mitochondria, the presequences are removed by several processing peptidases, including the mitochondrial processing peptidase (MPP), the inner membrane processing peptidase (IMP), the inter-membrane processing peptidase (MIP), and the mitochondrial rhomboid protease (Pcp1/PARL). Their proper functioning is essential for mitochondrial homeostasis as the disruption of any of them is lethal in yeast and severely impacts the lifespan and survival in humans. In this review, we focus on characterizing the structure, function, and substrate specificities of mitochondrial processing peptidases, as well as the connection of their malfunctions to severe human diseases.

## 1. Introduction

Mitochondria are vital components of all eukaryotes. They supply their cells with energy, maintain calcium homeostasis and biosynthesize heme and steroid molecules. They are also the main junction point of several metabolic and signaling pathways. Rather unsurprisingly, therefore, mitochondrial dysfunction is connected with an enormous range of diseases including myopathies (e.g., Kearns-Sayre syndrome), neurodegenerative diseases (e.g., Alzheimer’s disease (AD) and Parkinson’s disease (PD)), processes involved in ageing, and various types of cancers.

Generally, proper mitochondrial function depends on two sets of proteins. The smaller set is synthesized directly inside the mitochondria, while the larger set is synthesized on the cytosolic ribosomes. Cytosolically synthesized proteins have a presequence on their N-terminus that causes them to be imported into mitochondria using a complex transportation system [1,2,3,4,5,6,7,8]. After translation, cytosolic heat shock proteins chaperone the unfolded polypeptides to the outer mitochondrial membrane. Here, they are recognized by the receptors of the TOM (translocase of the outer membrane) complex and cross the outer membrane. In the intermembrane space, mitochondrial pre-proteins are immediately directed to the TIM23 (translocase of the inner membrane) complex for further translocation into the mitochondrial matrix (Figure 1) [9].

Outer mitochondrial membrane proteins and inner mitochondrial membrane carrier proteins possess an N-terminal, C-terminal and/or internal signal sequences instead of a cleavable presequence [10]. Upon mitochondrial localization, the N-terminal presequence must be removed to avoid problems with further sorting and protein folding and assembly. For this task, specialized metalloproteinases, mitochondrial processing peptidases, have evolved.

The mitochondrial processing peptidase (MPP) plays an essential role in this process: MPP deletion was shown to be lethal [11,12]. MPP is responsible for processing the protein precursors that are fully translocated in the mitochondrial matrix as well as the precursors in transit to the inner membrane or inter-membrane space (Figure 1). Several inter-membrane proteins possess a so called bipartite presequence consisting of both a mitochondrial targeting sequence (MTS) cleaved by MPP and also of an intermembrane space-sorting signal that is subsequently eliminated by the inner membrane peptidase (IMP) residing in the inner mitochondrial membrane (Figure 1) [11].

In addition, some proteins possess a typical octapeptide (Phe/Leu/Ile-XX-Ser/Thr/Gly-XXXX) in their targeting presequence and after cleavage by MPP, are further processed by the mitochondrial intermediate peptidase (MIP) located in the mitochondrial matrix [11]. Moreover, a rhomboid-type serine protease, Pcp1 in yeast and PARL in humans, was also shown to act in the processing of several mitochondrial inner membrane proteins [13]. The peptidases involved in presequence cleavage are quite conserved in eukaryotes, including yeast and humans, which reflects their vital role in mitochondrial biogenesis. In humans, mutations in these peptidases were reported to have an impact on the development of several serious diseases including neuropsychiatric disorders [14,15,16], Friedreich’s ataxia (FRDA) [17], autosomal recessive spinocerebellar ataxia type 2 (SCAR2) [18], and type 2 diabetes (T2D) [19] (see Table 1 below).

Here, we provide an overview of the mitochondrial processing peptidases, their functions, substrates, and the diseases associated with their mutations.

## 2. Mitochondrial Processing Peptidase

The principal responsibility of the mitochondrial processing peptidase is to remove the N-terminal targeting presequences of proteins imported into the mitochondria (Figure 1). MPP is a hetero-dimeric protein consisting of two subunits, α and β, which are referred to as PMPCA and PMPCB in humans (Figure 2A and Figure 3A,B) [11,12]. Yeast MPP is the only mitochondrial processing peptidase with a known crystal structure [20], although structures of several others have recently been predicted by AlphaFold [21]. These subunits together create a large substrate-binding cavity with a Zn^2+^-binding site on the MPPβ subunit (Figure 2B). Peptide bind cleavage is thought to occur through a reaction similar to that of thermolysin [20]: a water molecule complexed to the Zn^2+^ ion is polarized by nearby glutamate, thereby allowing it to carry out a nucleophilic attack on the carbonyl of the peptide bond. The Zn^2+^-binding site itself is created by a conserved HxxEHx_76_E motif in the MPPβ subunit (Figure 3B); the mutation of any of these residues eliminates Zn^2+^ binding and blocks the peptidase activity. Although the β subunit contains the entirety of the catalytic site, the cooperation of action of both MPP subunits is required for proper processing of pre-proteins.

The most conserved part of all known MPPα subunits is a glycine-rich loop (Figure 2A,C and Figure 3A) (GRL; residues G^284^GGGSFSAGGPGKGMYS^300^ in yeast MPPα), which is essential for substrate binding [23,24] and which moves the precursor protein towards the active site through a multistep process [25]. A crystal structure of an inactive yeast mutant MPP complexed to a peptide substrate showed that the peptide bound in an extended conformation in the active site (Figure 2D), forming a short series of β-sheet-like interactions with the β-sheets of MPPβ before proteolysis occurred [20]. An electrostatic analysis of the complex showed that the binding cavity was strongly negatively-charged while the substrate peptide is positively charged (Figure 2E,F).

MPP processes the N-terminal signal presequences of the majority of pre-proteins imported into the mitochondria. Although the presequences can vary in length and amino-acid composition, they have several common properties. They are all predicted to form an amphiphilic α-helix [26,27], have an overall positive charge, and have an arginine residue at position -2 or -3 from the cleavage site [28].

Besides, MPP has been shown responsible for the processing of precursor proteins that are post-translationally cleaved into polypeptides functioning as separate enzymes. For example, Arg5,6 from *S. cerevisiae* synthesized in the cytosol is after the import into mitochondria first cleaved by MPP to be disposed of its MTS and then internally processed to form two distinct enzymes, Arg5 (N-acetyl-gamma-glutamyl-phosphate reductase) and Arg6 (acetylglutamate kinase) [29].

Moreover, PMPCA has been reported to be SUMOylated at multiple sites [30,31]. Its SUMOylation was shown important for the processing of thioredoxin 2 (TRX2) in primary human umbilical vein endothelial cells (HUVECs). Such post-translationally modified MPP binds the SUMO interaction motif present in TRX2, which mediates its presequence processing. When the SUMO motif of MPP was mutated, TRX2 existed only as the precursor form, which was unable to protect cells from reactive oxygen species (ROS) generation and oxidative stress-induced senescence [32].

Deletion of both MPP encoding genes (MPPA and MPPB) is incompatible with the viability of *S. cerevisiae* under any and all growth conditions, including even anaerobic growth [33,34]. In humans, mutations in either PMPCA or PMPCB cause mitochondrial diseases that are characterized by neurological disorders with early childhood onset and a severe neurodegenerative course [35,36,37] (Figure 3A,B).

Jobling et al. [38] were the first to associate defects in PMPCA with human mitochondrial disease (Table 1). They identified mutations in PMPCA in 17 patients from four families affected with autosomal recessive non-progressive cerebellar ataxia: a homozygous missense mutation c.1129G>A (p.Ala377Thr), located in close proximity to the conserved glycine-rich loop, and the heterozygous mutations c.287C>T (p.Ser96Leu) and c.1543G>A (p.Gly515Arg), located in two other conserved regions (Figure 3A). They found that patients with these mutations had reduced levels of PMPCA and altered frataxin (FXN) processing. Frataxin is involved in the assembly of iron-sulfur clusters in the mitochondria [39] and its cleavage results in an intermediate form (amino acids 42–210) that is further processed into its mature isoform (amino acids 81–210). Patients’ cells showed an abnormal accumulation of the FXN42-210 isoform and decreased levels of the mature FXN81-210 isoform resulting in an increased mitochondrial oxidation/reduction ratio [40], which may occur due to the reduced electron flux in the respiratory chain [38]. The accumulation of unprocessed frataxin leads to the manifestation of non-progressive autosomal recessive spinocerebellar ataxia type 2, a neurological disorder characterized by the onset of impaired motor development and ataxic gait in early childhood. Brain imaging of affected individuals revealed cerebellar atrophy and the loss of fine motor skills, dysarthria, nystagmus, cerebellar signs, and delayed cognitive development with intellectual disability. Overall, the disorder is non- or slowly progressive, with patients typically surviving into adulthood [38].

**Figure 3 ijms-23-01297-f003:**
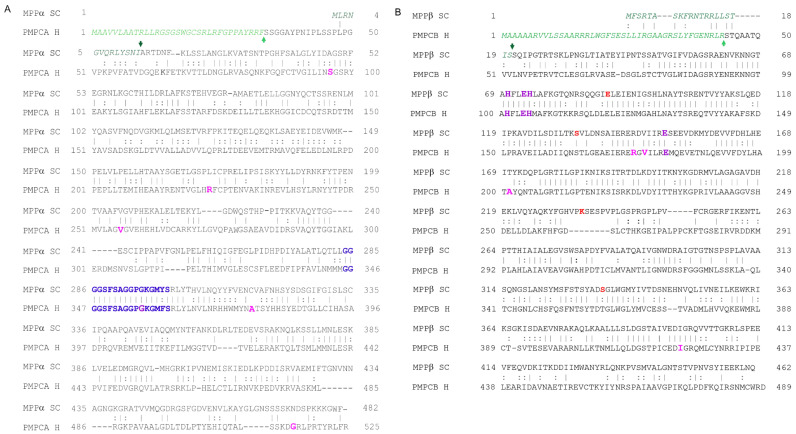
Amino-acid sequence alignment of MPPα and MPPβ from *S. cerevisiae* with PMPCA and PMPCB from humans. (**A**) Amino-acid sequence alignment of MPPα from *S. cerevisiae* (UniProt ID P11914) with human PMPCA (UniProt ID Q10713) performed by CLUSTALO [41]. The identity and similarity between both proteins are 31.3% and 44.9%. The presequence in all proteins is shown in green and italics and the presequence cleavage sites are marked with green arrows. The conserved glycine-rich loops, G284-S300 in MPPα and G335-S361 in PMPCA are colored violet and are in bold. The enlarged magenta residues of PMPCA are associated with human diseases (see the text above). (**B**) Amino-acid sequence alignment of MPPβ from *S. cerevisiae* (UniProt ID P10507) with human PMPCB (UniProt ID O75439). The identity and similarity between both proteins is 41.2% and 60.2%. The zinc finger motif present in MPPβ is colored violet. Residues whose mutation abolishes the catalytic activity are colored red. The enlarged magenta residues of PMPCB are associated with human diseases (again, see the text above).

Using a yeast two-hybrid assay, Koutnikova et al. [42] found that frataxin is also a direct partner of human PMPCB. A deficiency in mitochondrial levels of FXN causes Friedreich′s ataxia, an autosomal-recessive degenerative disease, which is associated with abnormal intramitochondrial iron handling [43,44]. Experiments suggested that two C-terminal missense mutations of FXN found in patients with FRDA (Gly127Val and Ile151Phe) modulate its interaction with PMPCB resulting in lower FXN processing efficiency which may contribute to its pathogenicity [42].

Recently, a novel heterozygous PMPCA variant c.677C>T (p.Arg223Cys) with a deletion of c.853del (p.Asp285Ilefs*16) was reported in a 15-year-old Japanese girl with severe and progressive developmental delay, cerebellar ataxia and extrapyramidal symptoms that seem to represent a new type of SCAR2 [18] (Table 1). Here, again, the deficiency of functional FXN results in an excessive mitochondrial iron accumulation, reduced iron-sulfur (Fe-S) cluster formation vital for the mitochondrial ATP production and increased oxidative damage [45]. The magnetic resonance imaging of her brain showed extensive cerebellar atrophy and excessive iron accumulation in deep grey matter nuclei [18].

Also, the brain imaging of a 7-year-old boy with global psychomotor delay, spastic-ataxic gait and upper limb dystonia showed cerebellar atrophy and an advancing hyperintensity in the striatum. A homozygous mutation in *PMPCA*, c.553C>T (p.Arg185Trp) was identified as manifesting with multiple defects of mitochondrial oxidative metabolism and signs of mitochondrial proliferation [46] (Table 1).

Another *PMPCA* homozygous missense mutation, c.766G>A (p.Val256Met), was identified in two siblings with cerebellar ataxia [47] (Table 1). Here, an immunoblot analysis did not reveal decreased levels of PMPCA, but, like the Ala377Thr mutation mentioned above, a consistent increase in the levels of the FXN42-210 isoform and a decrease in the levels of the FXN81-210 isoform was present. In contrast to the majority of patients described by Jobling et al. [38], these patients seem to have a milder course of disease without any intellectual disability [47].

In addition, a c.1066G>A (p.Gly356Ser) mutation in the glycine-rich loop (residues 345-358) of PMPCA together with the c.1129G>A (p.Ala377Thr) mutation, located 20 residues downstream of the loop, leads to a severe mitochondrial disease manifested by muscle weakness, extensive brain atrophy, visual impairment, and respiratory defects [48] (Table 1). Once again, the disease is associated with a reduction in PMPCA levels, abnormal frataxin processing and swollen mitochondria. Since MPP is responsible for the cleavage of imported subunits, its mutated form could lead to an incomplete processing of PMPCA itself and, consequently, further problems in assembling a functional MPP complex. Interestingly, the unassembled yeast MPPα was shown to be degraded by the mitochondrial ATP-dependent protease Lon [49].

Because it holds the catalytic site, the PMPCB subunit has a relatively mutation-intolerant gene, which is why PMPCB variants occur rather rarely. In 2018, however, Vögtle et al. [50] found that mutations to *PMPCB* were the primary cause of a mitochondrial disease resulting in Leigh-like neurodegeneration with very early and surprisingly strong impairment of iron-sulfur cluster biogenesis. The decreased activity or dysfunction of Fe-S cluster-containing enzymes lead to massive defects in crucial cellular functions. In patients, this manifests as a complex neurodegenerative condition with prominent cerebellar atrophy in early childhood. They reported several biallelic *PMPCB* variants (c.523C>T (p.Arg175Cys) with c.601G>C (p.Ala201Pro), c.524G>A (p.Arg175His) with c.530T>G (p.Val177Gly); and c.1265T>C (p.Ile422Thr)) (Table 1), all of which affect amino-acid residues that are highly conserved in yeast and humans (Figure 3B). To better understand the effect of these mutations on human MPP activity, they employed the crystal structure of yeast MPP. Although these mutations did not directly influence its catalytic activity, they may result in changes in PMPCB stability, protein folding, accessibility of the active site, or dimer formation.

Disruptions in mitochondrial metabolism are also typical for cancer cells. Hepatocellular carcinoma (HCC) tumors constantly evolve resistance to cytotoxic and targeted agents, resulting in failed treatment and tumor recurrence. Short hairpin RNA (shRNA) screening identified a transcript that conferred higher susceptibility to sorafenib, a multikinase inhibitor used to treat advanced HCC [51]; additional screening suggested that MPP could be a candidate and that silencing the PMPCB enhanced PINK1 (*PTEN*-induced putative kinase 1)-Parkin signaling and downregulated the anti-apoptotic protein MCL-1 (induced myeloid leukemia cell differentiation protein 1), thus sensitizing the HCC tumor cells to sorafenib therapy, potentially opening its use for the treatment of liver cancer [52].

**Table 1 ijms-23-01297-t001:** The overview of human mitochondrial processing peptidase mutations and their involvement in human diseases.

Processing Peptidase	Protein Variant	Disease, Symptoms	Ref.
**PMPCA**	Homozygous mutation: c.1129G>A (p.Ala377Thr) Heterozygous mutations: c.287C>T (p.Ser96Leu) with c.1543G>A (p.Gly515Arg)	SCAR2 with non- or slowly progressive cerebellar ataxia and developmental delay	[38]
	Homozygous mutation: c.766G>A (p.Val256Met)	slowly progressive SCAR2 without intellectual disability	[47]
	Heterozygous mutation: c.677C>T (p.Arg223Cys) with c.853del (p.Asp285Ilefs*16)	SCAR2 with progressive cerebellar ataxia and onset in infancy	[18]
	Heterozygous mutations: c.1066G>A (p.Gly356Ser) with c.1129G>A (p.Ala377Thr)	SCAR2 with progressive, extensive brain atrophy, muscle weakness, visual impairment, respiratory defects	[48]
	Homozygous mutation: c.553C>T (p.Arg185Thr)	SCAR2 with psychomotor delay	[46]
**PMPCB**	Heterozygous mutations: c.523C>T (p.Arg175Cys) with c.601G>C (p.Ala201Pro); c.524G>A (p.Arg175His) with c.530T>G (p.Val177Gly) Homozygous mutation: c.1265T>C (p.Ile422Thr)	Prominent cerebellar atrophy in early childhood	[50]
**IMMP2L**	Duplication: 46,XY,dup(7)(q22.1-q31.1)	GTS/TS	[15]
	Deletions ranged from ~49 kb to ~337 kb	Neurological disorders (ADHD, GTS/TS, OCD, ASD, Asperger′s syndrome, schizophrenia and developmental delay)	[53,54,55,56]
	Base pair change	Autism	[57]
	Copy number variation	Alzheimer′s disease	[58]
	Downregulation	Prostate cancer	[59]
**MIP**	Homozygous SNV: p.K343E Heterozygous SNVs: p.L582R with p.L71Q; p.E602* with p.L306 and p.H512D with 1.4-Mb deletion of 13q12.12	LVNC and developmental delay, seizures, hypotonia	[60]
	Heterozygous mutation: c.916C > T (p.Leu306Phe) with c.1970 + 2 T>A (p.Ala658Lysfs*38)	Developmental delay, hypotonia and intellectual disability	[61]
	Hypomethylation	Metabolic syndrome	[62]
	Downregulation	Prostate cancer	[59]
**PARL**	Reduced levels	Type 2 diabetes	[19]
	Leu262Val polymorphism	Increased plasma insulin concentration	[63]
	Mutation: c.230G>A (p.Ser77Asn)	Parkinson′s disease	[64]

## 3. Mitochondrial Inner Membrane Peptidase

The mitochondrial inner membrane peptidase is responsible for the maturation of proteins transported into the mitochondrial inter-membrane space (Figure 1) [65,66,67,68]. These include mature proteins synthesized both within the mitochondria (e.g., yeast mitochondrially encoded subunit 2 of cytochrome *c* oxidase, Cox2), or nuclear-encoded proteins synthesized in the cytosol and then transported into the mitochondria (e.g., yeast cytochrome b2, Cyb2, cytochrome c1, Cyt1, and NADH cytochrome b5 reductase, Mcr1).

Structurally, IMP consists of two subunits; in humans, these are IMMP1L (inner membrane mitochondrial peptidase 1-like) and IMMP2L (inner membrane mitochondrial peptidase 2-like) [66], and in *S. cerevisiae* there are three subunits, Imp1, Imp2, and Som1 [67]. Although the sequence identities between the individual yeast and human IMP homologues are relatively low (between 25–37%; Figure 4A), their tertiary structures, as predicted by AlphaFold [21,69], share a number of common features (Figure 4B–E). It should be noted here that the AlphaFold structures, both of IMP and those below, are only predicted structures and are to illustrate features that were originally derived from biochemical experiments or sequence analysis. All four IMP homologues are predicted to have a membrane-anchored α-helical N-terminal domain and a catalytic C-terminal domain. The yeast Imp1 and Imp2 subunits (Figure 4B,C) share 31% amino-acid sequence identity and both possess catalytic activity and are bound to the inner mitochondrial membrane [70,71,72]. The catalytic domain possesses a catalytic Ser/Lys dyad, which is present in all four proteins (Figure 4) and is structurally located in the C-terminal region [68,73]. The third yeast subunit, Som1, most likely serves to recognize substrates and was shown to physically interact with Imp1 [68,74]. Surprisingly, Som1 seems to be important for the Imp1-mediated proteolytic processing of Cox2 and Mcr1, but not for the maturation of the Cyb2 and Cyt1 cytochromes processed by the Imp2 subunit [68,74].

Two conserved glycine residues in the middle of the catalytic domain of both Imp1 and Imp2 subunits (Figure 4A) are necessary for either mutual subunit stabilization or overall proteolytic activity. Of these two residues, the glycine closest to the catalytic lysine appeared to be important for mutual subunit stability, while the one closer to the C-terminus is vital for proteolytic activity [67]. Interestingly, while the C-terminally located glycines are conserved in both human subunits (IMMP1L and IMMP2L), the glycine closest to the catalytic lysine appears only in IMMP1L but not IMMP2L (Figure 4A).

The currently known natural Imp1 substrates all possess a characteristic [I/V][H/D/F/M][N](↓)[D/E] amino-acid motif surrounding the cleavage site (indicated by ↓) [67]. Although the substrate specificities for Imp1 and Imp2 do not overlap, there are recognizable similarities between the protein precursors that they cleave. These include a hydrophobic residue at position -3 from the cleavage site and, for the nucleus-encoded substrates, the distances between the transmembrane segment and the cleavage site are also preserved. The accessibility of the cleavage site to the peptidase is also a prerequisite for cleavage by IMP [67].

In humans, the IMP homolog, IMMP2L (Figure 4E) has a 41% similarity to the yeast Imp2 subunit and a 90% similarity to the mouse IMMP2L [66]. It is composed of 175 amino acids with a gene of 860 kb located on chromosome 7q (AUTS1 locus), whose integrity has been shown to be critical for the development of autism spectrum disorders (ASDs). IMMP2L is expressed at a basal level in all human tissues except for the lungs and liver of adults [15,53].

Mutations associated with the gene encoding IMMP2L have been observed in several neurodegenerative diseases [54,75] (Table 1). In 2001, Petek et al. [15] were the first to link *IMMP2L* mutation to a childhood-onset neurobehavioral disorder called Gills de la Tourette syndrome or Tourette′s syndrome (GTS/TS). A boy with a de novo inverted duplication of a DNA segment on the long arm of chromosome 7 did not develop typical autistic symptoms but was observed to suffer from involuntary motor and vocal tics (typical GTS features), delayed speech development, depression, and several other conditions [15]. Likewise, intragenic microdeletions in *IMMP2L* were connected to a neurological disorder in patients aged 14 months to 9 years, and are thought to be a risk factor for the development of neurological diseases, including not only TS, but also attention-deficit hyperactivity disorder (ADHD), ASD, and schizophrenia [53,54,56,76]. Gimelli et al. [54] found *IMMP2L* microdeletions in four patients diagnosed with a neurological disorder. These patients showed dysmorphisms, hypotension, psychomotor and language delay, microcephaly and occasionally epilepsy. One patient showed some autistic symptoms, while two others showed mild skeletal defects [54]. Autism has also been observed in patients having two single-nucleotide base-pair changes in *IMMPL2* [57]. Furthermore, a copy number variation of the *IMMP2L* gene has also been observed in patients with late-onset Alzheimer’s disease [58].

The *IMMP2L* gene contains a neuronal leucine-rich repeat gene (*LRRN3*) located on its large third intron, which is highly expressed during fetal brain development. Studies in both *Drosophila* and mice demonstrated that the LRR family members could play an important role in various stages of neuronal development thus, making *LRRN3* another interesting candidate for autism [77]. Moreover, dysregulation of genes involved in central nervous system development has been observed in the knockdown of *IMMP2L* in human primary astrocytes [78]. However, Zhang et al. [55] did not confirm the association between *IMMP2L* deletions and the development of ASD, although they did find that the prevalence of *IMMP2L* deletions in patients was higher (3%) than in the negative controls (1.52%). Therefore, they believe that IMMP2L could play a role in autism spectrum disorders as an imprinting gene [55].

Similar to its yeast homolog, the IMMP2L subunit is responsible for cleaving the targeting sequence of cytochrome c1 and mitochondrial glycerol phosphate dehydrogenase 2 (GPD2) recognized by Imp1 in yeast [79]. In mice, mutations in both *IMMP2L* alleles cause an impairment in Cyt1 and GPD2 maturation [80]. Mouse mitochondria without *IMMP2L* were shown to produce an increasing amount of superoxide but showed normal GPD2 and mitochondrial respiratory complex III activities, as well as the normal level of mitochondrial bioenergetic capacity [81]. However, *IMMP2L* knockdown mice were observed to suffer from erectile dysfunction and oogenesis [80], bladder dysfunction [82], reduced food intake [83], early onset of ataxia and kyphosis [84], and age-dependent spermatogenic damage [80,84]. Liu et al. [85] suggested that the given symptoms are caused by the high production of superoxide in their mitochondria and its negation of nitric oxide (NO) acting as a signaling molecule and consequently, the increased production of other forms of ROS. As a result, ageing *IMMP2L* knockdown mice showed degeneration of cerebellar granule neurons, which could be treated with the antioxidant SkQ1 targeted to the mitochondria, which reduced the effects of oxidative stress in mice tissues [85]. A heterozygous mutation in mouse *IMMP2L* was associated with increased cases of infarction, increased ischemic brain damage and cerebellar granule neurons apoptosis [85,86].

IMMP2L can also act in cell fate determination, particularly in ”switching” between senescence and cell death, as the cell death regulator, and one of its substrates is AIF (apoptosis-inducing factor 1), a component of mitochondrial respiratory complex I [87]. In healthy cells, IMMP2L cleaves GPD2, which further participates in the normal signaling cascade; however, upon oxidative stress, IMMP2L processes AIF, which becomes active and triggers a signaling cascade that leads the damaged cell into apoptosis. Conversely, in cells that are “programmed” for the senescence process, the IMMP2L-GPD2 pathway is ”turned off”, resulting in non-functional phospholipid biosynthesis, reduced cell growth and further ageing. Senescence is associated with high levels of ROS. Thus, IMMP2L and GPD2 act as pro-survival proteins, while IMMP2L and AIF function proapoptically in cells damaged by oxidative stress. A shutdown of IMMP2L signaling is associated with the natural ageing process and protects against the premature apoptosis of senescent cells arising from the increased level of ROS in mitochondria that naturally occurs. According to these findings, IMMP2L could be a new biomarker for senescence and could contribute to healthy longevity [87].

## 4. Mitochondrial Intermediate Peptidase

The mitochondrial intermediate peptidase is important for the maturation of a subgroup of precursor proteins imported into the mitochondrial matrix or embedded into the mitochondrial inner membrane [43]. These pre-proteins are first processed by MPP and only afterwards by MIP, which cleaves an additional octapeptide following MPP cleavage. The cleavage site targeted by MIP is characterized by an RX(↓)(F/L/I)XX(T/S/G)XXXX(↓) motif [65] and is located at the C-terminus of a leader peptide (↓). Active MIP is a soluble monomer of 75 kDa in yeast and 81 kDa in humans (Figure 5). Its proteolytic activity is stimulated by manganese, magnesium and calcium ions while 1 mM Co^2+^, Fe^2+^ or Zn^2+^ completely inhibits it. Unlike MPP, MIP is also sensitive to N-ethylmaleimide (NEM) and other sulfhydryl reagents [88].

Positioning at the substrate N-terminus and a large hydrophobic residue (phenylalanine, leucine and isoleucine) at position -8 from the cleavage site are both essential features for cleavage by MIP; this type of substrate specificity is not shared by any other known peptidase [65].

In *S. cerevisiae*, mitochondrial oxidative phosphorylation is severely affected when *mip1* is missing. Branda et al. [65] showed that at least three vital components of the yeast mitochondrial gene expression machinery—mitochondrial small ribosomal subunit protein MrpS28, single-stranded DNA-binding protein Rim1, and elongation factor Tuf1—are processed by MIP. These proteins are essential for maintaining mitochondrial protein synthesis and mitochondrial DNA replication, which explains why the loss of *mip1* impairs the mitochondrially encoded OXPHOS subunits. *MIP1* disruption also results in the failure of at least two yeast nuclear-encoded respiratory chain components, the cytochrome *c* oxidase subunit 4 (Cox4) and the Rieske iron-sulfur protein of cytochrome bc_1_ catalytic subunit, to be cleaved [43].

In humans, MIP is encoded by the *MIPEP* gene, which contains 19 exons and is located on chromosome 13q12.12 [89]. *MIPEP* is expressed at high levels in energy-dependent tissues, such as the heart, brain, skeletal muscles, and pancreas [89,90,91]. Previously, some patients were reported with mutations in *MIPEP* which may have been linked to their diagnoses, but the first study showing that *MIPEP* is truly involved in a human disease was published in 2016 by Eldomery et al. [60] (Table 1). They identified several single nucleotide variants (SNVs) in the *MIPEP* gene that caused a loss of MIP function in four unrelated patients suffering from an oxidative phosphorylation deficiency. All four children presented left ventricular non-compaction (LVNC), delayed development, seizures, hypotonia, cataracts, and infantile or early childhood death caused by cardiomyopathy [60]. A similar study described a patient with a heterozygous *MIPEP* mutation, who also had a developmental delay, severe intellectual disability and hypotonia, however, unlike the previous ones, did not suffer from cardiomyopathy at the age of 20 [61] (Table 1). Further studies in human fibroblasts showed that MIP has an important role in OXPHOS function since its loss impaired the processing of several OXPHOS subunits, including the OXPHOS complexes I, IV and V [61]. The mitochondrial inner membrane protein, OXA1L, which is involved in the proper assembly of OXPHOS complexes I, IV and V, was also found to be a MIP substrate [92,93]. In patients with metabolic syndrome, a cluster of conditions that increase the risk of cardiovascular diseases, diabetes type II and cancer morbidity and mortality, *MIPEP* was also found to be one of the most hypomethylated genes in white adipose tissue; this led to an increased expression of MIP [62]. In non-metastatic prostate cancer, 11 genes associated with mitochondrial integrity and function were downregulated, including both *MIPEP* and *IMMP2L*. Decreased *IMMP2L* expression was also correlated with cancer-related fatigue in prostate cancer patients receiving radiotherapy, which suggests that impairment of mitochondrial function may be involved in the development of radiation-induced fatigue [59]. Finally, Pawlowski et al. [94] described five genes with the most significant expression changes in malignant canine mammary tumors from the lowest to the highest grade of tumor malignancy. One of them was *MIPEP*, which had increased expression in grade III tumors and the lowest expression in grade I tumors; it could thus serve as a potential marker of mammary malignancy in dogs [94].

## 5. Mitochondrial Rhomboid Protease

The mitochondrial rhomboid protease—Pcp1 (processing of cytochrome *c* peroxidase protein 1) in yeast and PARL (presenilin-associated rhomboid-like) in humans—plays an essential role in mitochondrial quality control and steady-state maintenance [13,95,96]. Pcp1/PARL is a member of the rhomboid family of intramembrane serine proteases, which have a core consisting of transmembrane helices (Figure 6). Preliminary cellular localization studies demonstrated that the N-terminal part of PARL is localized in the mitochondrial matrix, while its C-terminus extends into the mitochondrial intermembrane space. Conserved serine and histidine catalytic residues (Ser256/His313 in Pcp1, and Ser277/His335 in PARL; Figure 6A) are responsible for its intramembrane proteolytic activity [97].

While there is no significant conservation in the N-terminal regions of yeast Pcp1 and human PARL (Figure 6A), this region is strongly conserved among vertebrates, especially in mammals [98]. The N-terminal domain of PARL is processed by two cleavages (Figure 6A,C) [99]. The first is mediated by MPP, which removes the mitochondrial targeting sequence, while the second is developmentally regulated and depends on PARL’s own intramembrane-cleaving protease activity. The PARLΔ53 truncated form (without its MTS) is considered to be its mature form and is predominantly present in lung, brain, heart and muscle tissues [97]. The second cleavage occurs at Ser77 and releases the so-called Pβ peptide, whose sequence is conserved only in mammals. This Pβ peptide is exported into the nucleus and contributes to mammalian-specific, developmentally regulated signaling between the mitochondria and nucleus [99]. Phosphorylation of PARL has also been proposed to play a key role in regulating its activity. Ser65, Thr69, and Ser70 have all been identified as possible PARL phosphorylation sites: phosphomimetic mutations at these sites dramatically reduced Pβ-cleavage and the PARLΔ77 production [100]. As a result of that, phosphorylation increases PARLΔ53 levels, which enhances its activity towards substrates [101].

The C-terminal parts of Pcp1 and PARL have higher sequence identity, where several transmembrane α-helices are located (Figure 6A–C). Six of them are present in both Pcp1 and PARL, while PARL has an additional transmembrane α-helix (H4 in Figure 6A). Both proteins possess endopeptidase activity and their catalytic dyads, Ser256 and His313 in Pcp1 and Ser277 and His335 in PARL, are highly conserved and both occur in the transmembrane helices in their respective proteases (Figure 6A).

Yeast Pcp1 is known to process two mitochondrial substrates, cytochrome *c* peroxidase (Ccp1), and mitochondrial genome maintenance protein 1 (Mgm1), a mitochondrial outer membrane dynamin-like GTPase [102]. Both substrates are processed in two steps. For Ccp1, its MTS initially directs the import of the Ccp1 pre-protein until its transmembrane segment is localized in the inner mitochondrial membrane. Subsequently, a heme is bound to apo-Ccp1, causing it to fold and become resistant to proteases acting in the intermembrane space. The transmembrane segment of Ccp1 is then further translocated and removed by *m*-AAA, and subsequently processed by the intramembrane proteolytic activity of Pcp1. The second yeast Pcp1 substrate, Mgm1, was shown to exist in two different lengths, both of which are required for its proper functioning [103]. The large Mgm1 isoform, which contains an N-terminal transmembrane segment, is processed by MPP after import into the mitochondria. This functional isoform is an integral inner membrane protein facing the intermembrane space. Part of it is later processed by Pcp1 and loses its anchoring transmembrane domain. Yeasts lacking Pcp1 were shown to be defective in Mgm1 processing, leading to increased levels of the full-length protein and reduced levels of its shorter isoform [103,104]. A Pcp1-deficient yeast strain was unable to grow in a glycerol medium and displayed striking mitochondrial morphological abnormalities, such as mitochondrial fragmentation, aggregation, and loss of mitochondrial DNA [105].

Mammalian PARL is responsible for the processing of several mitochondrial substrates, e.g., the serine protease HTRA2 (high-temperature requirement factor A2), the serine/threonine-protein kinase PINK1, the serine/threonine-protein phosphatase PGAM5 (phosphoglycerate mutase family member 5), and OPA1 (optic atrophy protein 1), the human homolog of yeast Mgm1 [13]. Full-length HTRA2 localizes to the inner mitochondrial membrane, whereas PARL-processed/activated HTRA2 is released into the intermembrane space [106]. Processed HTRA2 regulates apoptosis in human lymphocytes by preventing the accumulation of the pro-apoptotic regulator Bax in an active form. PARL is thus implicated in regulating the survival of lymphocytes and neurons. An in vivo study investigating the role of PARL and HTRA2 in mouse striatal neuronal injury occurring after transient global brain ischemia found a parallel decrease in neuronal PARL expression and processed HTRA2 levels, consistent with the possibility that HTRA2 could be a pathophysiologically relevant PARL substrate in these stress conditions. Moreover, PARL downregulation through siRNA silencing significantly worsened the outcome of transient ischemia [107].

**Figure 6 ijms-23-01297-f006:**
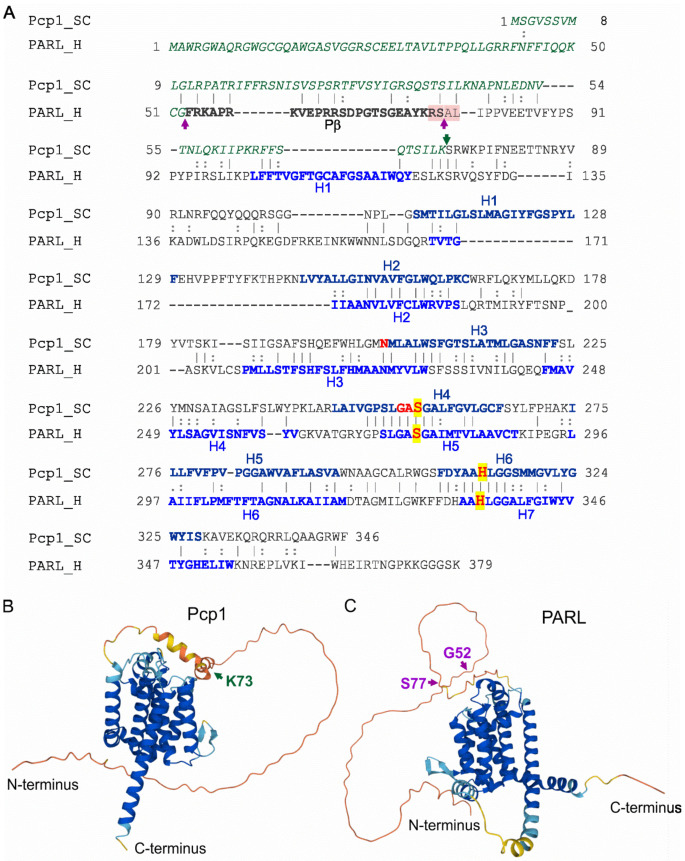
Amino-acid sequence alignment and structural comparison of yeast Pcp1 and human PARL models. (**A**). Amino-acid sequence alignment of mitochondrial rhomboid protein 1 (Pcp1) from *S. cerevisiae* (UniProt ID P53259) and human PARL (UniProt ID Q9H300) performed by EMBOSS Needle [108]. The overall sequence identity and similarity are 21.3% and 33.7%, respectively. The mitochondrial transit peptides of Pcp1 (residues 1–73) and PARL (residues 1–52) are shown in green and italics. Pβ of PARL (residues 53–77) is shown in black bold. The unique cleavage sites of Pcp1 and PARL are marked with purple and green arrows, respectively (the prediction of the actual structures surrounding these residues is uncertain, but very likely they are intrinsically disordered). Residues 76–79 of PARL whose mutations abolish the Pβ cleavage are highlighted light pink. Blue bold residues mark the Pcp1 and PARL transmembrane α-helices, H1–6 in Pcp1 and H1–7 in PARL. S256 and H313 of Pcp1 and S277 and H335 of PARL (highlighted yellow) are the catalytically active residues. The red residues: N202, GAS 254-256 and H313 of Pcp1 are crucial for maintaining its endopeptidase activity. (**B**,**C**). Molecular models of rhomboid proteins Pcp1 (**B**) and PARL (**C**). Pcp1 and PARL cleavage sites for transit peptides (K73 and G52) and Pβ (S77) are indicated by arrows. Coloring corresponds to the likelihood of the elements of the secondary structures: red—very low, yellow—low, sky blue—confident, deep blue—very confident. The structure models were prepared by AlphaFold [21,69] and were cross-referenced in the UniProt database.

In functional mitochondria with an intact electrochemical potential, full-length 63 kDa PINK1 is imported into the mitochondria through the outer and inner membrane translocases, the TOM and TIM complexes. In the matrix, the mitochondrial targeting sequence is removed by MPP, and the remaining 60 kDa protein is inserted into the inner mitochondrial membrane [109]. Membrane-docked PINK1 is then rapidly cleaved between position Ala-103 and Phe-104 in its transmembrane domain by PARL to generate a 53 kDa form, which relocates to the cytosol [110]; cytosolic PINK1 is rapidly degraded by the proteasome following the N-end rule pathway [111]. A large mitochondrial accumulation of PINK1 is observed upon the downregulation or genetic activation of PARL [110,112,113].

Unlike PINK1, the processing of the serine/threonine-protein phosphatase PGAM5 is increased when the mitochondrial potential has collapsed, which shows that it has a role in stress response and is regulated in an opposite way by PARL [114]. The overexpression and downregulation of PARL also substantially influences the levels of PGAM5 [101].

By analogy with yeast Mgm1, OPA1 should be a substrate for PARL, but several other proteases also operate in producing mature OPA1 forms and it is thought that PARL cleaves OPA1 only under specific physiological or pathophysiological conditions [105]. It has also been suggested that OPA1 and PARL interact in a non-enzymatic manner [13].

Recent proteomic analyses identified several additional PARL substrates including a subunit of mitochondrial respiratory chain complex III, TTC19 (tetratricopeptide repeat domain 19), the pro-apoptotic protein Smac (second mitochondrial-derived activator of caspases), the mitochondrial lipid transferase STARD7 (StAR-related lipid transfer protein 7), and CLPB (caseinolytic peptidase B), a putative mitochondrial chaperone. The identification of these substrates further supports the role of PARL in mitochondrial homeostasis [101,115,116,117].

The membrane localization of PARL assumes its interactions with lipids located in the inner mitochondrial membrane (IMM). In humans, PARL interacts with cardiolipin, a lipid exclusive to the IMM of eukaryotic cells. Cardiolipin represents approximately 10 % of total cellular lipids and is essential for the activity of numerous IMM proteins. Also, it significantly increases the in vitro activity of PARL, which could either enhance the protease stability or contribute to the conformational changes supporting the substrate binding or substrate entrance to the PARL active site [118].

The phenotype of *Parl*^−/−^ or PARL-deficient mice is characterized by progressive multisystemic atrophy from their fourth postnatal week, eventually resulting in cachectic death [119]. A more detailed study found that in *Parl*^−/−^ fibroblasts and myoblasts, and in purified *Parl*^−/−^ liver mitochondria, enhanced mitochondrial network remodeling and the mobilization of cristae stores of cytochrome *c* occurred, suggesting that PARL and OPA1 could be part of the same molecular pathway of death [119].

Reduced PARL levels have also been connected with type 2 diabetes (T2D) and ageing (Table 1). Civitarese et al. [19] showed that lowering PARL levels in human muscle cells resulted in lower mitochondrial oxidative capacity, reduced mitochondrial mass, increased protein oxidation and ROS production and impaired insulin signaling, all of which are known metabolic defects in T2D and ageing. Moreover, screening performed on 1031 North Americans found that the presence of a Leu262Val polymorphism in PARL was associated with increased plasma insulin concentration, making it a risk factor for diabetes. Interestingly, two other groups [63,120] were unable to confirm this observation when screening the populations of the UK and Ireland, though one of them, Hatunic et al. [63] did find that the Leu262Val PARL genetic variation is associated with an earlier onset of diabetes and may be a marker of increased susceptibility to nephropathy and cardiovascular complications in patients with diabetes.

An additional PARL missense mutation, Ser77Asn, was detected in two patients diagnosed with Parkinson’s disease, which also implicates PARL dysregulation in PD pathogenesis [64] (Table 1). This mutation is located in the highly conserved N-terminus of the protein, in a position that is crucial for the second maturation of PARL and the release of the Pβ peptide (see above and Figure 6). Unfortunately, further independent studies failed to detect this mutation or any other pathogenic mutation in the *PARL* gene during the screening of patients with both sporadic and familial PD. This suggests that if *PARL* mutations are a genetic cause of Parkinson′s disease, then they are likely to be extremely rare [121,122].

## 6. Conclusions

Mitochondria are semiautonomous organelles that use a sophisticated system for importing pre-proteins synthetized in the cytosol and directing them to their final destination within mitochondria. This system is necessary for maintaining proper mitochondrial function and homeostasis. Mitochondrial processing peptidases are important components of the mitochondrial import machinery; their responsibility lies in removing the signal presequences of peptides and pre-proteins transported into the mitochondria. Most of these pre-proteins are processed by the mitochondrial processing peptidase, MPP, while other processing peptidases, including the mitochondrial inner membrane peptidase, IMP, the mitochondrial intermediate peptidase, MIP, and the mitochondrial rhomboid proteases, Pcp1/PARL, are responsible for the further processing of specific protein substrates. Disruption of any of these proteases is lethal, and their malfunctions are connected with a number of severe human diseases. These diseases are characterized by an early onset and patients often exhibit neurodegeneration. The short peptides produced by these proteases might also accumulate inside the organelle, causing further problems; consequently, functional coupling between the processing of precursor proteins and presequence degradation is crucial for maintaining a functional organellar proteome [123]. The mitochondria eliminate these peptides in two ways. Either the free presequences can be exported into the cytosol, like the Pβ peptide of PARL, which is further translocated into the nucleus where it associates with chromatin and contributes to developmentally regulated mitochondria-to-nuclei signaling, or, like the majority, they can be degraded by a peptidase to generate amino acids that can be used for the synthesis of new proteins within the mitochondria. In humans, the mutation of these peptidases may also lead to severe diseases.

## Figures and Tables

**Figure 1 ijms-23-01297-f001:**
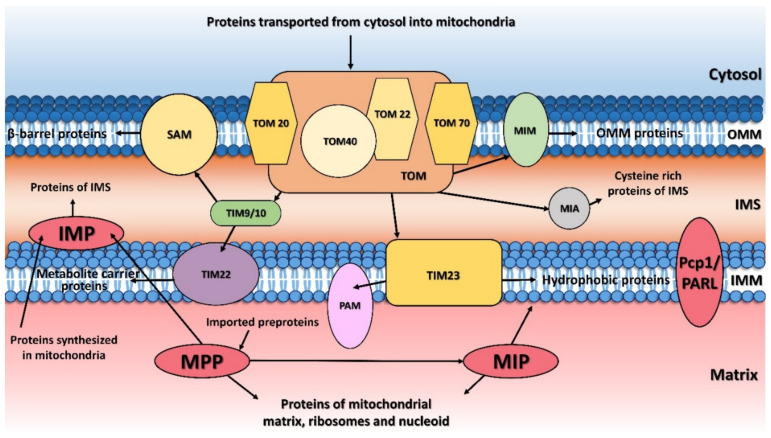
Protein transport and processing from cytosol to mitochondria. MTS (mitochondrial targeting sequence)-carrying pre-proteins are imported through the TOM and TIM23 complexes. Proteins containing hydrophobic sorting signal are embedded into the inner membrane (IMM), while hydrophilic proteins are sent into the mitochondrial matrix through the PAM (protein import motor) complex. Cysteine-rich proteins are imported by the TOM and MIA protein translocation systems. The precursors of β-barrel proteins are translocated through the TOM and TIM9/10 complexes and sorted and assembled by the SAM complex. Metabolite carrier precursors are imported via TOM, TIM9/10 and TIM22, and several α-helical outer mitochondrial membrane (OMM) proteins are imported by the MIM complex. Pre-proteins imported into the mitochondria are processed by the mitochondrial processing peptidase (MPP) and later by the mitochondrial intermembrane peptidase (IMP) or mitochondrial intermediate peptidase (MIP). Proteins synthesized inside the mitochondria themselves are processed by IMP. Some inner membrane proteins are processed by rhomboid protease Pcp1/PARL. OMM, mitochondrial outer membrane; IMM, mitochondrial inner membrane; IMS, intermembrane space.

**Figure 2 ijms-23-01297-f002:**
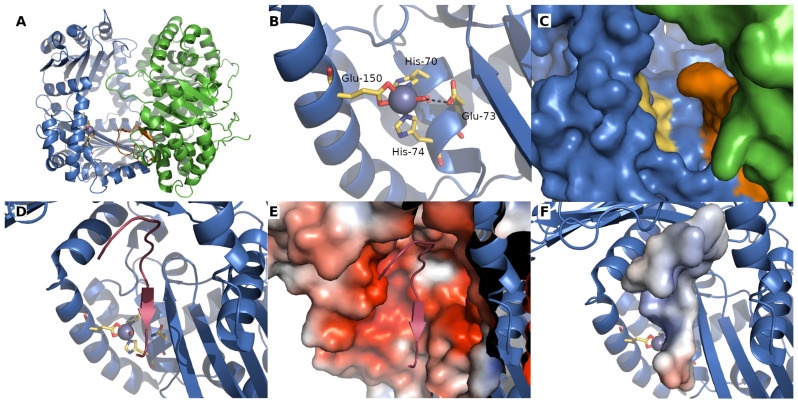
Structure of yeast MPP. (**A**) MPP forms a dimeric complex consisting of α (green) and β (blue) subunits. The Zn²⁺-binding site residues found in the MPPβ subunit are shown as sticks and colored yellow, while the glycine-rich loop, which is important for substrate binding, is colored orange. (**B**) The Zn²⁺-binding site consists of His-70, Glu-73, His-74, and Glu-150. The water molecule bound to the Zn²⁺ ion is thought to be polarized by Glu-73 in order to allow it to carry out a nucleophilic attack on the carbonyl of the peptide bond during substrate hydrolysis using a reaction mechanism similar to that of thermolysin (see [20]). (**C**) The glycine-rich loop (orange) controls access to the catalytic site (yellow). (**D**) The signal peptide of cytochrome C oxidase IV (dark red) is bound to an E73Q mutant of the MPP complex. The substrate binds in an extended conformation, forming main-chain β-strand type hydrogen bonds with residues 101–104 of MPPβ. (**E**) The active-site cavity where the substrate binds is generally strongly negatively charged, while the substrate itself is generally positively charged (**F**). The electrostatic surfaces in (**E**,**F**) were calculated using APBS [22] and all images were created using PyMOL 2.5; panels (**A**–**C**) show yeast wild-type MPP (PDB ID 1HR6), panels (**D**–**F**) show the yeast MPP E73Q mutant (PDB ID 1HR8).

**Figure 4 ijms-23-01297-f004:**
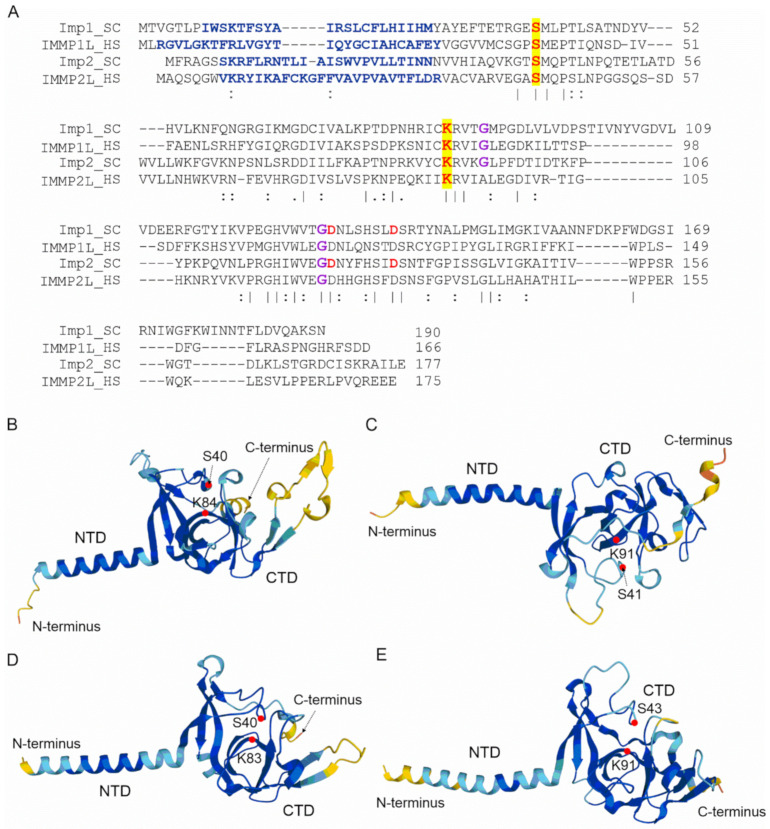
Amino-acid sequence alignment and structural comparison of *S. cerevisiae* Imp1 and Imp2 with the human IMMP1L and IMMP2L proteins. (**A**). Amino-acid sequence alignment of *S. cerevisiae* Imp1 (UniProt ID P28627) and Imp2 (UniProt ID P46972) with the human IMMP1L (UniProt ID Q96LU5) and IMMP2L (UniProt ID Q96T52) proteins performed by CLUSTALO [41]. Sequence identities between pairs range from 25–37%, though the overall identity is over the whole multiple sequence alignment and only 10.95%. The N-residues forming the N-terminal transmembrane α-helix are colored blue; the catalytic serine and lysine residues of all four proteins are colored red and highlighted in yellow. Those glycines, which are important for the stability and for maintaining the endopeptidase activity in yeast IMP are colored violet. Red aspartic acid residues are crucial for maintaining endopeptidase activity (**B**–**E**). Comparison of predicted structural models of Imp1 (**B**) and Imp2 (**C**) from *S. cerevisiae* with the human IMMP1L (**D**) and IMMP2L (**E**). NTD, CTD indicate the N- and C-terminal domains, while the Ser/Lys catalytic dyads are marked with red circles. The coloring corresponds to the predicted likelihood of the elements of the secondary structures: red—very low, yellow—low, sky blue—confident, deep blue—very confident. These structures were prepared by AlphaFold [21,69] and cross-referenced in the UniProt database.

**Figure 5 ijms-23-01297-f005:**
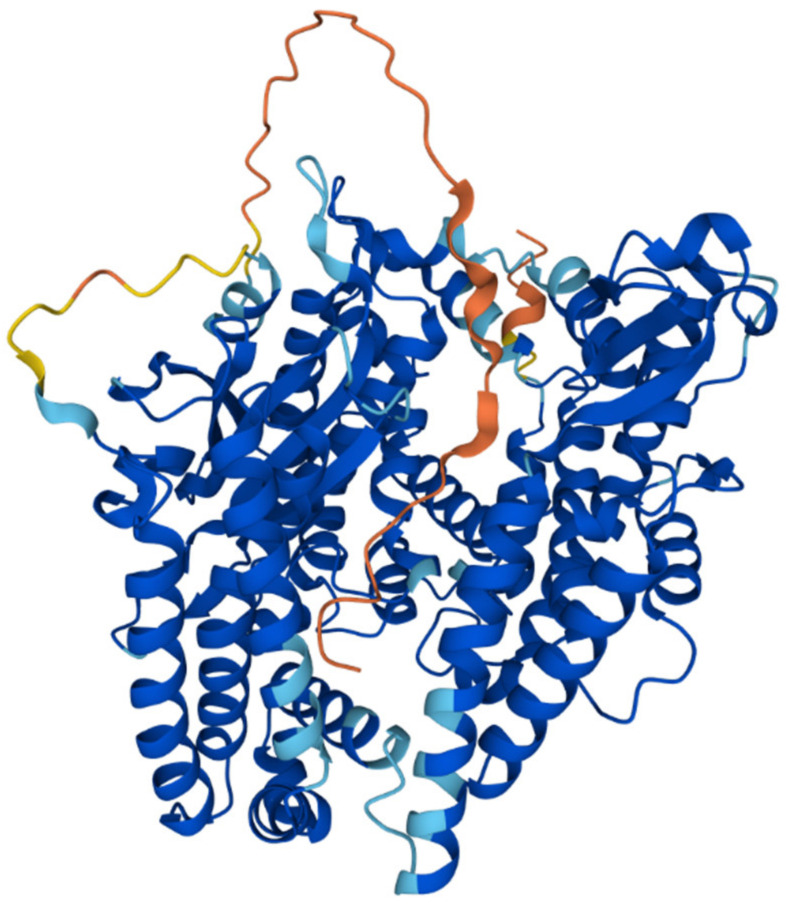
AlphaFold model structure of the human mitochondrial intermediate peptidase MIP (UniProt ID Q99797)**.** The coloring corresponds to the predicted likelihood of the elements of the secondary structures: red—very low, yellow—low, sky blue—confident, deep blue—very confident. The structure was prepared by AlphaFold [21,69] and was cross-referenced in the UniProt database.

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
