# Peer review of "Mitochondrial Processing Peptidases—Structure, Function and the Role in Human Diseases"

_ijms, 2022, doi:10.3390/ijms23031297_

Round 1

Reviewer 1 Report

This is a comprehensive and well-organized review article that will give researchers a better understanding of the roles of the mitochondrial processing peptidases and theirs’ malfunctions are connected

with a number of severe human diseases.  The work will be a significant contribution to this field.  I have only one suggestion that if possible, making a Table to list the up-to-date research, in which the mutations of MPP, IMP, MIP, and Pcp1/PARL associated with human diseases.

Author Response

We would like to thank the reviewer for a positive feedback on our manuscript. We have addressed to his/her suggestion on providing the table of mentioned human diseases. We hope that the revisions based on the comments of the second reviewer will be acceptable.

Reviewer 2 Report

The submitted manuscript entitled “Mitochondrial processing peptidases and their role in human diseases” is a review article focused on discussion of the latest advancements in the field of structure, functions, and the involvement in mitochondrial disease initiation and progression of mitochondrial processing peptidases. The topic can be of interest to the audience of a special issue of the journal and the manuscript is well-written. However, there are some concerns and recommendations that can help to improve the quality of the manuscript.

Major concerns:

  1. Since large parts of the manuscript focus on structure, functional sites, and there is little information on human diseases it is recommended to change the title of the manuscript. Consequently, the Abstract should be re-considered.
  2. The Figure 3 legend states that the figure was created using PyMOL This is a protein structure visualization system that uses either experimentally obtained structures from the PDB database or obtained through 3D structure modeling. Which structures were visualized here? Also, the legends of Figures 4, 5, and 6 stated that the figures were prepared by AlphaFold. Here, (i) the authors should indicate the Uniprot IDs that were used to model the 3D structures; (ii) Were the obtained 3D models validated? (iii) Alpha-Fold can give errors in the obtained structures and intramolecular interactions and, therefore, the analysis of functional sites can be wrong.

Minor concerns:

  1. It is recommended to (i) shortly discuss general principles of functioning of MPPs such as pre-sequence proteolysis upon a protein organelle import and sorting; () emphasize in the manuscript that MMPs belong to metallopeptidases.
  2. Introduction, line 64 – what is a sequence of the octapeptide?
  3. The mechanisms underlying the involvement of MPPs in disease development should be clarified throughout the manuscript. For example, a sentence on lines 189-190 should be clarified and a mechanism is the enhanced PINK1-Parkin signaling and downregulated the anti-apoptotic protein MCL-1.
  4. From 109 Refs, only 16 are from the last 5 years – 2017 to 2021. More recent papers should be discussed.

Author Response

We would like to thank the reviewer for a positive feedback on our manuscript and valuable comments and suggestions for further improvements. We have tried to address to all of his/her points, which are stated it the enclosed PDF file.

Round 2

Reviewer 2 Report

The authors properly addressed all my concerns and have made corrections to the manuscript.